# Stigma and its impact on disclosure and mental health secrecy in young people with clinical depression symptoms: A qualitative analysis

**Katie Prizeman**◯*, **Ciara McCabe, Netta Weinstein**

Department of Psychology and Clinical Language Sciences, University of Reading, Reading, United Kingdom

* katiediab@gmail.com

**Data Availability Statement:** Prizeman, Katie (2023): Data supporting 'Stigma and its impact on disclosure and mental health secrecy in young people with clinical depression symptoms: a

## Abstract

### Background

Clinical depression ranks as a leading cause of disease and disability in young people worldwide, but it is widely stigmatized. The aim of this qualitative research was to gather young people's experiences of depression stigma and its impact on loneliness, social isolation, and mental health disclosure and secrecy. This novel information can then be used to guide psychosocial interventions for young people with depression.

### Methods

This qualitative study included $N = 28$ young people aged 18–25 years ($M_{age} = 21.30$). Participants were recruited from the community who had high symptoms of depression (assessed through a pre-screen using the Mood and Feelings Questionnaire (MFQ) with a benchmark score > 27) or had been recently diagnosed with depression by a medical professional. Semi-structured interviews were based on conceptual model drawings created by participants and analyzed using thematic analysis.

### Results

Four main themes emerged: 1) *Depression secrecy: positive and negative aspects*; 2) *Depression disclosure: positive and negative aspects*; 3) *The solution is selective disclosure*; and 4) *Participants' recommendations do not align with personal preferences*. In particular, the young people described non-disclosure as a way to be in control, but that secrecy prevented authentic engagement with others. Young people also described disclosure as eliciting more stigma but as necessary to gain help. Finally, the young people described struggling with knowing how much to disclose in relation to their mental health and with whom they could disclose.

qualitative analysis'. University of Reading. Dataset.
https://doi.org/10.17864/1947.000517.

**Competing interests:** The authors have declared that no competing interests exist.

**Abbreviations:** MDD, Major depressive disorder; MFQ, Mood and Feelings Questionnaire; TA, Thematic analysis.

## Conclusions

This study provides new evidence of how young people with depression experience stigma and its effects on disclosure and mental health secrecy. Knowing how young people struggle with these issues can allow us to develop interventions to encourage them to come forward and discuss their mental health in order to receive appropriate support and treatment. We recommend young people be signposted and have access to mental health champions or nominated teachers in their schools or universities.

## Introduction

According to the World Health Organization, clinical depression, also known as major depressive disorder (MDD), ranks as a main cause of disease and disability in young people across the globe. Young people are more prone to depression compared to other age groups [1, 2], with depression onset extending from mid-adolescence to the mid-40s [3]. However, over 40% of individuals experience their first episode of clinical depression before the age of 20 [3].

With past research describing a bidirectional link between loneliness and depression, young people also have a higher prevalence of loneliness [1]. Loneliness is a negative emotional state noticed when there is a perceived gap between wanted and real social connections [4]. This subjective construct is frequently associated with others referring to disrupted relationships, such as social isolation [5], alienation [6], deficits in social connectedness [7], a reduced sense of belonging [8], and deficits in social capital [5]. Research has shown the detrimental effects of loneliness on physical health [5, 9]. But a recent meta-analysis shows that loneliness has medium to large negative effects on all health outcomes, with the largest effects on mental health and overall well-being [10].

One possible determinant of loneliness in depressed young people is mental health stigma [11]. Stigma is a broad concept comprising negative stereotypes, prejudices (endorsement of stereotypes and emotional reactions), and discriminatory behaviours against people with mental conditions [12–14]. Mental health stigma takes two forms: 1) public stigma; and 2) internalized stigma. Public stigma is when members of society discriminate against people with mental conditions, whereas internalized stigma is when individuals with mental conditions internalize negative stereotypes and expect labels to be applicable to their diagnosed illness [15, 16].

Young people with depression often have stigma associated with their depression symptoms [14, 17, 18]; there is reason to believe stigma undermines well-being and exacerbates loneliness [11]. Outside the population of depressed young people, stigma has been linked to loneliness, isolation, and social withdrawal (i.e., reduced social networks, interpersonal relationships, or a sense of belonging in the community) [19, 20].

Young people in particular may be even more sensitive to stigmatizing experiences [11, 21]. The transitional period from childhood to adulthood is a sensitive one in which lived experiences can have long-term effects on future health and development [22, 23]. For example, stigma seems to play a detrimental role in the well-being and help-seeking of young people with depression [16, 21, 24]. Stigma is also considered to be one of the main reasons for experiencing loneliness among people with mental illness [25]. The most obvious way in which stigma breaks a person's social ties is through rejection by others and discrimination in many areas of life [26].

In anticipation of being rejected, people with mental health issues might also avoid social contact [13, 27]. Past research has found that internalized stigma can affect people's social networks, personal relationships [17], and feelings around being part of a community [28]. As a result of stigma, individuals may struggle to decide whether or not to disclose their mental health issues to minimize the risk of further stigma and labeling [29]. Secrecy may shield people from additional stigma temporarily, but it typically has long-term negative effects such as loneliness, isolation, and poorer relationship experiences [20, 30, 31]. While disclosure carries the risk of being stigmatized and labeled by others, it is thought to have less long-term detrimental effects, including internalized stigma [11, 12, 31, 32].

### Current study

Little is known about how mental health stigma might affect young people's decisions to disclose their depression [33–35] and the implications of this on loneliness and isolation in young people [36, 37]. This study was developed to reveal how stigma impacts depression, secrecy, and disclosure, as well as its effects on loneliness and social isolation.

To address research gaps, this study aimed to answer two research questions:

RQ #1: What are the subjective views of mental health stigma among young people with depression symptoms?

RQ #2: What influence does stigma have on loneliness, social isolation, secrecy, and disclosure of depression?

## Methods

### Study design

The present study involved a qualitative methodology. Following an interpretative approach [38], semi-structured interviews were used to develop an understanding of young people's views of public and internalized stigma and its effect on secrecy and disclosure of depression, as well as feelings of loneliness and social isolation. An overview of the qualitative interview guide is presented in Table 1.

### Participants and recruitment

Participants' demographics and clinical characteristics are presented in Table 2.

Participants included $N = 28$ young people aged 18–25 ($M = 21.30$) and were recruited from the community utilizing advertisements and posters between March 10th, 2023, and April 20th, 2023. After completing socio-demographic and a depression questionnaire, the Mood and Feelings Questionnaire (MFQ), semi-structured interviews were then scheduled and conducted with participants by KP between March 21st, 2023, and April 20th, 2023 (post-COVID-19 lockdown). The study strictly followed the standards of voluntary and informed consent and data protection for each participant who voluntarily fulfilled the inclusion criteria. Young people between the ages of 16 and 25 who had high scores (>27) on the MFQ [39]. Some participants had also been diagnosed with clinical depression. That said, we did not collect this data on diagnosis. Nor did we collect data on participant psychiatric medication. Of the $N = 33$ participants who initially volunteered to take part, only $N = 28$ were included due to depression score inclusion criteria (>27). There were no other inclusion or exclusion criteria.

### Ethical considerations

This study was conducted in accordance with the Declaration of Helsinki. The study proposal was approved by the University Research Ethics Committee (2022-072-NW) of the University

**Table 1. Qualitative interview guide.**

| Focus area | Sample questions and probes from interview protocol |
|---|---|
| **Socio-demographic questions:** | 1. Age:<br>2. Country:<br>3. Education level (School/University):<br>4. Ethnicity:<br>5. Gender (Male/Female/Other/Prefer not to say): |
| **Stigma's impact on the disclosure and secrecy of depression:** | 1. Could you please explain the links you have made in your conceptual model drawing and the reasons why?<br>2. Do you think stigma has a large impact on you when disclosing or keeping your mental health condition a secret? Please explain and give reasons for your answer.<br>3. What are the reasons, if any, for keeping your mental health condition a secret? Please explain why.<br>4. What are the reasons for you to disclose your mental health? Please explain why.<br>5. Who do you choose to talk to, and why? Please explain your answer. How do these people react when you tell them about your mental health condition? How does this make you feel?<br>How does the experience of stigma impact your relationships with your family? How about with your friendships?<br>6. Who do you not want to talk to, and why not?<br>How does it feel to hide your mental health? Does it make the stigma worse or bring about additional psychological pain?<br>7. Do you find talking about your mental health condition helpful? Please explain your answer.<br>8. Does talking or not talking about your mental health condition have an impact on your feelings of loneliness and social isolation? Please explain your answer.<br>9. Have you ever talked about your mental health online?<br>If yes, how has being online changed your experience of mental health stigma?<br>10. Is there anything else you would like to share? |

of Reading on June 6th, 2022. Information about the study was given to the participants, including instructions on the nature of the study, their right to decline to answer any questions they wish, their right to withdraw, and data handling.

Written informed consent was sought for each participant who volunteered and fulfilled the inclusion criteria. Participants received a debriefing form upon completion of the survey. They were also given the option to be contacted about further studies, and their anonymized data would be publicly shared in a University of Reading repository for other researchers if requested.

## Data collection

Participants completed demographic questions about age, gender, education, and ethnicity.

## Mood and Feelings Questionnaire (MFQ)

Participants completed the Mood and Feelings Questionnaire (MFQ) [39] (high scores indicate greater depression symptomatology) as a pre-screening for depressive symptoms before taking part in the study. The MFQ is a 33-item scale that measures depressive symptoms in children and young adults. It has good psychometric properties [40, 41]. Participants' responses indicate how they have been feeling or acting in the past two weeks [39]. A cut-off score of 27 and above has been identified as the difference between clinical and non-clinical levels of depressive symptoms [42]. Each item is rated on a three-point Likert scale from 0, not

**Table 2. Participant demographics and clinical characteristics (N = 28).**

| Participant | Age[a] | Gender | Ethnicity | Country | Education level | MFQ scores (/66)[b] |
|---|---|---|---|---|---|---|
| P01 | 25 | Male | Hispanic/Latino/Spanish | United States of America | University | 62 |
| P02 | 18 | Female | White | United Kingdom | High school | 52 |
| P03 | 20 | Other | White | United Kingdom | University | 56 |
| P04 | 24 | Female | Asian | United Kingdom | University | 28 |
| P05 | 25 | Male | Black/African American | United Kingdom | University | 30 |
| P06 | 22 | Male | Black/African American | South Africa | University | 30 |
| P07 | 21 | Female | White | United Kingdom | University | 54 |
| P08 | 24 | Female | Arabic/Other racial-ethnic group | United Kingdom | High school | 53 |
| P09 | 19 | Female | Asian | United Kingdom | University | 51 |
| P10 | 19 | Female | White | United Kingdom | University | 59 |
| P11 | 21 | Female | White | United Kingdom | University | 56 |
| P12 | 25 | Female | Asian | United Kingdom | University | 28 |
| P13 | 18 | Female | White | United Kingdom | High school | 39 |
| P14 | 18 | Female | Asian | India | High school | 28 |
| P15 | 21 | Female | Asian | United Kingdom | University | 50 |
| P16 | 23 | Male | Black/African American | United Kingdom | University | 65 |
| P17 | 21 | Female | White | United Kingdom | University | 48 |
| P18 | 20 | Male | White | South Africa | University | 63 |
| P19 | 21 | Female | White | United Kingdom | University | 36 |
| P20 | 21 | Female | Arabic | United Kingdom | University | 40 |
| P21 | 22 | Female | White | South Africa | University | 46 |
| P22 | 23 | Female | Black/African American | India | University | 54 |
| P23 | 20 | Female | White | United Kingdom | University | 31 |
| P24 | 20 | Female | White | United Kingdom | University | 28 |
| P25 | 19 | Female | Black/African American | United Kingdom | University | 42 |
| P26 | 21 | Male | Asian | Malaysia | University | 28 |
| P27 | 20 | Male | White | United Kingdom | University | 35 |
| P28 | 25 | Male | Asian | United Kingdom | University | 28 |

*MFQ* Mood and Feelings Questionnaire (higher scores indicate more depression). All participants completed the long version of the MFQ.

[a]Age at interview

[b]MFQ score at screening or diagnosis

true, to 2, true. This questionnaire is widely used to score depression in young people, with higher scores suggesting more depression symptomatology [42].

## Conceptual model drawings

As a launching point for the semi-structured interview, each participant first developed a conceptual model drawing. This technique, devised by the authors, allowed participants to link research constructs in a visual form and use their output to discuss how and why they felt the constructs connected. The first author (KP) presented each participant with the following list of concepts with definitions and examples: mental health stigma—public stigma and internalized stigma; MDD or depression; secrecy; self-disclosure; social isolation; loneliness; and self-esteem. Participants were asked to create their own conceptual model drawing of how they think these concepts link together based on subjective experiences and views, and then explain how and why they have made these connections.

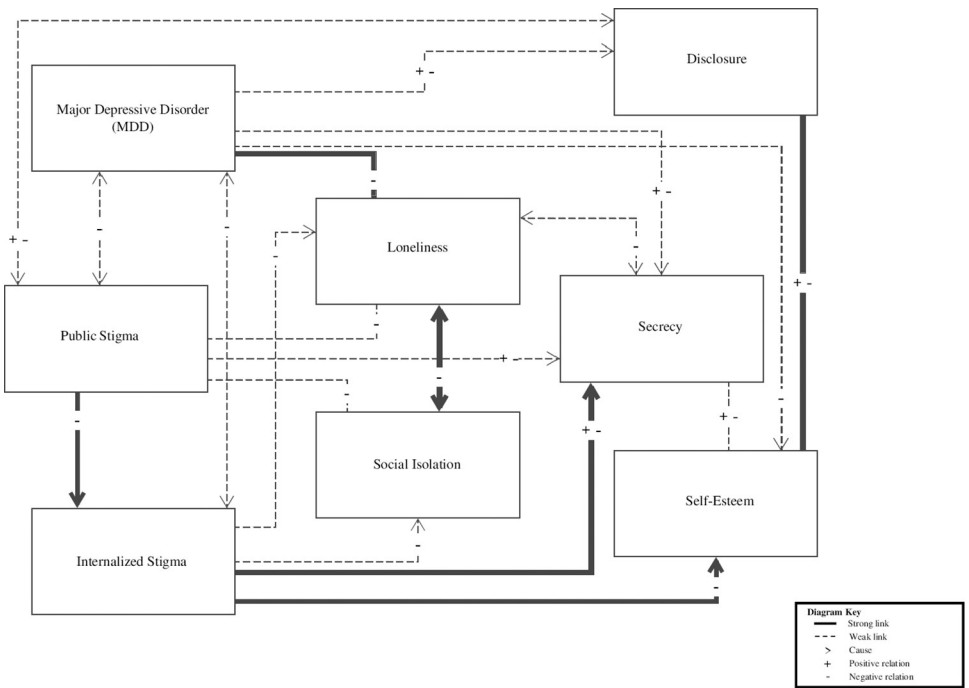

**Fig 1. Summary of participants' conceptual model drawings.**

KP then gave participants instructions on drawing the conceptual model. Instructions included: use all concepts provided; link as many concepts to one another as one feels necessary; link concepts together by connecting them with lines (use a thicker line to link concepts together to show a greater connection, and a thinner line to show a connection but not as strong as the thicker lines); include arrows to indicate a causal relationship between concepts. All drawings were based on participants' subjective feelings and experiences. Following that, semi-structured interviews discussed the way in which participants had linked the concepts. See Fig 1 for a summary of the participants' conceptual model drawings. Fig 1 shows a summary of participants' link research constructs in a visual form (thicker lines indicate a greater connection). This output was used to discuss how and why participants felt the constructs were connected. We did not interpret the summary model in the study but used it as the basis for further discussion with participants about the links they identified (described below).

## Semi-structured interviews

Semi-structured interviews with *N* = 28 young people were scheduled and conducted with participants by KP between March 21st, 2023, and April 20th, 2023 (post-COVID-19 lockdown). All interviews took place online (using the Microsoft Teams platform). Interviews were conducted until data saturation was reached, meaning that no new information was observed and collected [43]. Interviews were audio-recorded, transcribed verbatim, checked for accuracy, and subjected to thematic coding by KP. Each interview lasted between approximately 25 and 60 minutes (average 40 minutes) and was conducted in English.

A semi-structured interview format was chosen to give young people a voice. This technique also allowed participants to guide the direction of the interview based on their conceptual model drawings while remaining flexible to allow follow-up on noteworthy areas of accounts that may surface during the interviews. Follow-up prompts included: *"Can you tell*

*me more about this*?*" "Could you give me an example of this or any personal experiences related to you*?*"* were also used.

Discussing mental health stigma can distress participants, so in the final part of the interview, participants were given the opportunity to talk about any other opinions or experiences they would like to share to help improve their mood and end the interview on a positive note. Refer to Table 1 for an overview of the qualitative interview guide.

## Data analysis

Anonymized transcripts were analyzed using NVivo software, a qualitative data analysis software. Analysis was undertaken by the first author (KP) using a combination of inductive and deductive approaches [44–46]. KP independently reviewed each transcript using the inductive technique, then used the open coding method to extract and code significant data units [47].

The next step involved the research team (KP, CM, and NW) identifying and analyzing patterns of meaning in the dataset using thematic analysis (TA). This technique was chosen as it is ideal for investigating how a group conceptualizes a particular phenomenon [48–50].

TA is not tied to a particular ontological or epistemological position; therefore, in this study, the researchers adopted a post-positivist critical realist stance [51]. This position assumes that reality is observable and quantifiable while recognizing that participants are unaware of all the factors that influence their experience [48].

To create a codebook, initial codes were transformed into higher-order concepts and themes based on their shared characteristics. As part of the deductive methodology, the research team thoroughly discussed and evaluated major emergent themes while comparing new data with previously collected data (pre-existing codes were used as a guide and template for the clustering of newly collected data). The codebook was continually refined until data saturation was reached, at which point no new themes could be developed based on the data [43]. The researchers considered their own sources of bias and prior assumptions, including knowledge of depression and mental health stigma (KP, CM, and NW), when conducting research into young people's mental health (KP, CM, and NW).

## Results

Participants' demographics and clinical characteristics are presented in Table 2. All participants had an MFQ of <27. A cut-off score of 27 and above has been identified as the difference between clinical and non-clinical levels of depressive symptoms [42].

Four interrelated broad themes emerged from the data. Themes were used to explore research questions 1–2, which aimed to better understand the subjective views of stigma among young people with depression symptoms and the influence it has on loneliness, social isolation, secrecy, and disclosure of depression. The overview of themes and sub-themes can be seen in Table 3. To ensure the standard usage of English is kept, minimally corrected verbatim quotes are shown below.

## Theme one: Depression secrecy: Positive and negative aspects

This theme that emerged is directly related to both our research questions as it explores young people's decision to keep their depression a secret as a result of previous stigma experiences.

The majority of participants expressed stigma as a primary reason for their non-disclosure of depression. Recurrently, participant interviews showed the decision to keep depression a secret as an important and consequential reaction to previous subjective experiences and views of stigma.

Table 3. Table of themes and sub-themes.

| Themes | Sub-themes |
|---|---|
| Depression secrecy: positive and negative aspects | a. Non-disclosure as a way to be in control: the protective mechanism<br>b. Depression secrecy as preventing authentic engagement<br>c. Stigma's secrecy creates a vicious cycle |
| Depression disclosure: positive and negative aspects | a. Finding safety in disclosure: "the healing process"<br>b. Disclosure invites danger |
| The solution is selective disclosure | a. Know how much to disclose<br>b. Disclosure raises questions |
| Participants´ recommendations do not align with personal preferences | |

Despite individuals' past experiences and views of stigma, choosing secrecy showed beneficial effects on individuals' experiences, protecting individuals from stigma, rejection, and lowered self-esteem.

**Subtheme: Non-disclosure as a way to be in control: The protective mechanism.** Participants voiced their non-disclosure of depression as a way to feel in control of their interpersonal settings. They expressed a persistent desire or need for control that may have defended them against possible judgments from others, particularly in terms of the perception that they were unable to manage their lives because of depression. Control was a helpful tool participants used to help portray an image of themselves that they wanted others to see. Some were more explicit in stating that control meant having the power over who to share their depression with. Secrecy was a way for participants to create a sense of safety, security, and predictability.

"*Yeah, it's more about it's, it's pretty. . .it's all about control, and I feel like there's a big idea that that people with like mental health issues don't have self-control.*" (P27, 20, Male)

"*You don't want to share anything because you even in turn realize these ideas that there will be negative effects if you do or people will view you in these ways if you do and. . .Yeah, yeah. You want to control how you portray yourself so that others can see you in a way that is not only your depression.*" (P27, 20, Male)

"*. . .it feels like it gives me control over it or control, at least over. . .Like, I don't want people to to look at me as someone who cannot have self-control or is the way they are because of the depression.*" (P27, 20, Male)

In addition to being mentioned in many participant interviews as a strategy to maintain control, non-disclosure was also described as a function to protect self-esteem from unfavorable views by others that were then internalized into the self. Participants described that talking can lead to more harm than good, leading them to feel worse about themselves.

"*And and even if, even if someone is like me and doesn't particularly want to share you that just because you have a a diagnosis doesn't mean you then have to share it as well. You still have control over who you do and don't share to. . .And you know, that can have a big impact on your self-esteem.*" (P27, 20, Male)

"*Starting with self-esteem, I think that's the easiest one to explain. I linked that to internalized stigma and public stigma. . .Like they cause sort of lack of. . . Um really affect self-esteem. And*

*then you know, depending on how much self-esteem you have will probably affect your self-disclosure and how much you talk to others about, you know. . .Personal sort of things. And that sort of also relates to how, you know, secretive you're going to be."* (P13, 18, Female)

*"And there's there are other times when I think that the amount that you self-disclose to people would also kind of be dependent on your self-esteem, because if you are somebody who has a higher self-esteem, you might not find it an issue to self-disclose about your problems because it wouldn't affect how other people perceive it."* (P04, 24, Female)

Though secrecy had a protective function, it also had a detrimental impact on young people's experiences. In fact, considering the broader context, the personal and interpersonal consequences of stigma were largely caused by the way it encouraged secrecy surrounding one's depression.

**Subtheme: Depression secrecy as preventing authentic engagement.** Many participants were reluctant to disclose their depression for varying recurring reasons, such as feelings of non-acceptance by others, unauthentic social engagement, and feelings of abnormality and indifference. This led to frequent social withdrawal, along with added secrecy and stigma. Feelings of non-acceptance as a result of earlier views of depression's stigma were repeatedly expressed as a big reason why participants did not disclose their depression.

It was evident in their recounts that participants yearned for others' acceptance. However, participants expressed concerns that if they were open about their depression, their true selves might not be accepted by others. Participants felt that by keeping their depression to themselves, they were able to gain more social acceptance (i.e., their social selves were more accepted by others). Secrecy, although not always chosen, was the more desirable approach. It was a useful tool to gain acceptance from others by hiding aspects of the self.

*"Um, I think like public stigma is like a main, uh, bubble, which like connects to internalized stigma, secrecy, and social isolation. Because I think like public perceptions of things really does like affect how people view themselves, and it does like kind of form a big narrative on how things are like valued or perceived in society. So, I think it can cause like a lot of, um, secrecy, as people don't want, um, disclose like any mental health problems. And um, which in turn can cause social isolation as people with those mental health problems might not feel like they could be understood or accepted for who they are, so therefore might withdraw themselves from society."* (P09, 19, Female)

*"I think that internalized stigma can sometimes lead to secrecy, where you don't sort of tell people about your problems, as well as the mental health stigma and the public stigma. So, if you are constantly being told that you are abnormal, if you're going through something like a mental illness, or if you're suffering from it, then on the next occasion when you meet a fresh set of people, you might be tempted to keep that a secret just to be accepted."* (P04, 24, Female)

Few participants further described their non-disclosure of depression in terms of filtering out information and altering narratives as a desire to be accepted by others. This validated participant experiences of added stigma being directly linked to depression's stigma; for instance, others' stereotypical attitudes and beliefs about taking antidepressants or being hospitalized.

*"Because if you keep your mental health a secret, it can feel it can lay really heavy on your head, cause it's obviously such a massive part of everyone's lives that you know you are keeping a massive part of your secret from the people around you, and when you can just freely even mention it in a conversation. . .Like for example, if I talk to someone about my day, you*

*know, and if I have a counseling appointment that day, you know, if I know they're not going to be accepting, I'd have to leave that bit out. . .You don't feel accepted by them. And you know, you might get asked sort of like questions, and you know, just people putting in their input when it's not needed, I suppose."* (P13, 18, Female)

In addition, some participants suggested that stigma's secrecy relates to more superficial and ineffective relationships. In particular, participants recognized that by hiding their depression from those around them, they were withholding a large portion of their lives. This inevitably results in one not feeling ´completely known´ by others.

*"I also think that I do understand, to some level, that I do need more shallow friendships in my life. And friendships, you know, where I see them on a night out or I just see them in a seminar, and I wouldn't, you know, disclose all of this to them because I don't think that's the space that they hold in my life. So. . .and I don't think I know enough about them that I know that I wouldn't be judged."* (P20, 21, Female)

Overall, a key conclusion of this research was that hiding parts of the true self ultimately results in the continuation of a vicious cycle. For example, secrecy led to ineffective and superficial relationships, which exacerbated feelings of loneliness, social isolation, and low self-esteem. This process played a detrimental role in individuals' recovery (i.e., help-seeking and receiving support).

*". . .because if you feel like you can't be your true self around someone. . .It does definitely close. . .It breaks down relationships, cause it feels like they don't have the effort to truly like truly understand you. Cause it leads to secrecy in everything, and that's just that's not a good foundation for a relationship, which I guess can lead into loneliness. Yeah, more loneliness, but I also do think it leads to more social isolation too, cause if you're doing behaviours that you don't want someone to know about, I guess that feeds into social isolation."* (P11, 21, Female)

**Subtheme: Stigma's secrecy creates a vicious cycle.**   Participants described that as they experienced more stigma (either public or internalized), they kept more secrets about their depression. This resulted in a vicious cycle. For example, the more public stigma there is, the more it is internalized (i.e., participants saw internalized stigma as a direct outcome of public stigma), and the more feelings of loneliness and social isolation there are, the more secrecy there is.

*"Social isolation can be caused by loneliness, because if people feel like if people feel lonely surrounded by a group of people and they can't help that they feel lonely. . .they may be more of a tendency to draw themselves away from their social group. So that in turn is them distancing themselves from their group of friends. . .that in turn can cause secrecy because they´re keeping it to themselves. They´re keeping themselves to themselves. They may not want to socialize um with people for their own reasons, and yeah, this can make you feel very left out."* (P02, 18, Female)

*"Yeah, I think that hiding a lot about yourself. . .So it goes both ways, so hiding a lot about yourself has a massive impact on your self-esteem. . .not being able to talk about stuff. And it makes you kind of feel ashamed around that. And by not talking about it, you're then perpetuating that belief that something is wrong with you, that you shouldn't talk about, which will lower your self-esteem but also. . .So then also having low self-esteem around your depression*

*or whatever makes it a lot harder to be able to talk about things, and therefore you're much more likely to be secretive about um the problems you're facing, and that can be very lonely at times."* (P03, 20, Other)

"*Yeah, I've definitely not spoken to people for a while. Just for no real well, they think no reason but. . .Yeah, I definitely think that as soon as I feel lonely, the social isolation and the secrecy and everything like that definitely comes into play for me."* (P10, 19, Female)

"*Mental health stigma. . .internalized stigma and public stigma um cause loneliness because, you know, you feel like you can't talk to other people about um mental health and just in general talking to people, and then the internalized stigma that you've got from general mental health stigma you know, also causes you to just not talk to people about things. So you sort of do, in a way, cut yourself off from others, which will cause you to feel lonely."* (P13, 18, Female)

### Theme two: Depression disclosure: Positive and negative aspects

This theme that emerged is directly related to both our research questions in that it explores young people's decisions, reasons, benefits, and consequences of disclosing one's depression as a result of previous stigma experiences. It further highlights stigma's impact on young people's feelings of loneliness, social isolation, and lowered self-esteem.

Participants also discussed those moments when they chose to disclose their depression. Such disclosure served a purpose and had benefits. However, a key finding from this research was that disclosing one's depression also had a harmful effect on people's experiences. The dangers of stigma are mostly caused by the disclosure of depression. Legitimate reasons for the functions and barriers of disclosure are based on participant interviews and are discussed below.

**Subtheme: Finding safety in disclosure: "The healing process".** It was important for participants disclosing to feel comfortable and understood by others. Participants said they were more willing to share when they felt less initial stigma and judgement. Participants also found disclosing their depression helpful. For some, it led to feeling accepted and secure and receiving understanding and support from friends and family. In this way, disclosure acted as a source of healing. Healing occurred for reasons such as disclosure, which fosters freedom, reduces feelings of loneliness and social isolation, and increases empowerment as a way to break the stigma.

A few participants expressed feeling free after talking, despite others' lack of understanding or support. Participants felt better after disclosing, describing a weight being lifted from their shoulders.

"*It makes you open and feel free after talking."* (P16, 23, Male)

"*And it's being comfortable to disclose around people who you know have an understanding. And once you've gained, whether someone has that stigma or doesn't have that stigma it, then you can kind of tell whether it's safe or not to disclose that information."* (P03, 20, Other)

"*I think cause I've started speaking about it more and I've got like more positive reactions, I think it's reduced my need to wanna be secret like secret about like mental health, and it's reduced that loneliness feeling. Because I´m like ´Ohh the people out there get it. They also understand. ´ Yeah."* (P15, 21, Female)

That said, participants recognized that not everyone would always understand what they were going through, making it difficult and less likely for them to disclose their mental health

problems. Still, some participants remained open to sharing if others demonstrated a willingness to learn or some sort of understanding.

"*Yeah, I think I would sort of keep it a secret from somebody who doesn't understand, like my parents, who would always see it as a reflection on them and something that they did wrong in raising me, whereas I would be more comfortable talking about it with, let's say, like my boyfriend or my friends who have also at some point struggled with some kind of depressive episode or have been to therapy. And they understand that it's a it's normal. And everyone goes through it, so it can be part of the conversation.*" (P06, 22, Male)

"*Because they listen, they're willing to learn. I think that's another thing they're willing to learn. If they don't understand something, they will ask ´What?´ or ´Why do you do this or? ´. . . .A huge factor, a huge factor if you are willing to learn, then you feel more open and comfortable to talk about it. I'm less likely to talk to someone if they don't get it, it's really difficult to explain, but if that they're not open to learning, they're not open to trying to understand why someone does something the way they do.*" (P11, 21, Female)

Furthermore, one of the key reasons given by participants for disclosing their depression was to combat the effects that the stigma had on them. Participants understood that removing any existing stigmas about oneself was the first step toward recovery (both public and internalized). As a result, participants felt empowered to overcome stigma, with the goal of alleviating feelings of loneliness and social isolation.

"*The stigma makes you want to keep it secret. But the stigma also makes you want to self-disclose in order to break the stigma.*" (P11, 21, Female)

"*. . .it's like overcoming the stigma for yourself is like the most important part. So as soon as, like, there's sometimes people that I haven't wanted to tell and then once I have, it does feel a lot better. So, I think it is just kind of like a mental barrier sometimes you do have to, like get past. Yeah, I definitely think if you overcome like the internal stigma, it becomes easier cause once you talk about it, it kind of people around you learn, and then sometimes, like that person, would like the stigma that they have kind of decreases. And yeah, the talking would help you feel less lonely. So. . .I do think that overcoming the internal one can help.*" (P17, 21, Female)

"*I think so because once you're able to talk about it with someone, you're much less likely to feel that loneliness and isolation because you don't feel so alone with what you're feeling.*" (P19, 21, Female)

**Subtheme: Disclosure invites danger.**    Participants were sensitive to others' non-verbal communication, including their facial expressions, body language, and unspoken responses towards individuals disclosing their depression. Participants recognized others' discomfort as unhelpful and humiliating (i.e., others' discomfort increased participants' feelings of loneliness, social isolation, shame, and guilt). Instead of the support and understanding desired, participants were given adverse reactions and unsympathetic responses. As a result, some participants viewed disclosure as more harmful than helpful, increasing stigma, which in turn increased feelings of loneliness and social isolation and lowered self-esteem.

"*I think a lot of people when you. Um. Disclosure about any kind of mental health or depression. It makes people uncomfortable. And this can be. . . Um. It can be awkward and a bit embarrassing. Um. And therefore they, a lot of people won't make the effort trying to*

*understand better, in that can create more misunderstanding and make public stigma a lot worse."* (P03, 20, Other)

"*My depression made me lonely because doesn't feel like people understood what I was going through. Even the people I talked to about it later on, after I became a bit better, it still felt like I was in my own world and I was alone the whole time. Nobody understood. Nobody knew or people. . .it always felt like people didn't care."* (P18, 20, Male)

For many, non-verbal cues were interpreted as a motivator to disclose their mental health or not. One participant said she had no trouble telling someone about her depression as long as she received a positive response in return.

". . .*You sort of have to pick and choose, I suppose, who you talk to about it, depending on how you think they'll react and what they'll say. You´re more likely to share if you think they're going to be like accepting, and not judging, and being sort of willing to listen and learn. Um. If you think you´re sort of going to get a bad reaction off of them and they´ll think of you differently, or you see they are uncomfortable. . .Um, yeah you´re less likely."* (P13, 18, Female)

"*Yeah, but if they were fine with it, you know, and and I have no I have no problem sort of talking about it. It's just if I know the push back, I'd get with talking about it then I'll probably leave it out."* (P13, 18, Female)

That being said, some participants described having mental health difficulties that were debilitating and therefore required disclosure, though it may not have been their choice. Participants adapted in those circumstances to safe disclosure by selecting disclosure as necessary.

"*Yeah, I think that if you're around the right people talking about it helps because I think that with things like depression or agoraphobia or just kind of those types of things where you like your functioning is quite impaired and you know it doesn´t make a difference if you tell people about it or not because, um, it´s already quite obvious. Um, you don´t really have a choice. And you know, having people around you that can kind of celebrate your small wins and not put you on the same standards as other people really helps.*" (P20, 21, Female)

One participant described using others' overall support and reactions as a cue for whether to maintain those relationships. She added that, even though it might not be her choice, she needed disclosure in order to live a life of any quality.

"*I think the reason is because my mental health issues are quite debilitating in regards to like the probably depression slightly less so. But I also have OCD and agoraphobia and that kind of almost has to be disclosed for me to have any kind of quality of life. So, I feel like. . .Probably had they not been had have. . .if they weren't as debilitating as they were, I may feel less of a need to disclose them and because of the stigma and what I think people might think. But I think also because I have dealt with mental health issues since I was a very small child, I kind of learned that I can't really be happy in any capacity if I surround myself with people who, if they knew what I was experiencing, they wouldn't accept me.*" (P20, 21, Female)

"*. . .So, I kind of find it easier to just kind of like rip the bandage off and let people know that you know `This is difficult for me.´ So if it makes me feel worse about myself and they make me feel like a problem, I know to step away so. Yeah. So yeah, so I think that it it definitely does affect me, but I think I'm at a point where I kind of don't surround myself in situations where it would affect me as much as it would.*" (P20, 21, Female)

### Theme three: The solution is selective disclosure

This theme that emerged is directly related to our research questions in that it provides a better understanding of young people's stigmatizing subjective views and their awareness of how much to disclose to others. That is, to get the necessary support while alleviating addition stigma, judgmental reactions, and misunderstanding.

In response to the stigmas that surround depression, participants described being selectively open about their conditions. Selective disclosure had a valuable effect on individuals' experiences. The benefits of selective disclosure were drawn from knowing how much to disclose and how to explain depression during disclosures.

**Subtheme: Know how much to disclose.** Some participants talked about their family and friends' stigmatizing attitudes and perspectives, as well as their unhelpful or unsupported replies, when specifics regarding their depression were revealed. For example, they discussed judgmental reactions to their engaging in behaviours such as self-harm or hospitalizations that were associated with depression. They felt those carried an even greater stigma. Some found others' reactions to lack responsiveness, understanding, or to be insensitive, which resulted in increased feelings of stigma and discrimination, discouragement, and lower self-esteem. Participants shared an understanding of what information to communicate about their diagnosis and how much to divulge after observing others' reactions.

*"If you talk it out, you can open your heart out, and you feel more at peace and at peace with yourself other than if you continue to bottle it all in. You um just have to know how much to say."* (P01, 25, Male)

*"I think the only thing is that there are different levels of and things that you can disclose that have different kind of levels of stigma. Um. For example, if I were to say to someone, ´Ohh from time to time I experienced depression, ´ I feel like that has less stigma than if I was to kind of fully self-disclose and be like, `I've had depression since I was young. I've been hospitalized, ´ that kind of thing. They have very different kind of levels of stigma in the way I approach that. It's very different."* (P03, 20, Other)

*". . .because I think if other people make you feel bad about yourself, then you don´t necessarily come and open up to other people about it."* (P17, 21, Female)

*"There's definitely that fear of what other people are gonna think of you that fear of potentially losing people that might be close to you because they don't you know, you feel like they might not want to help you or might think badly of you. I think that's definitely one of the biggest things. If you think that or if you've already got a sort of opinion of mental health yourself, you might feel like telling people about what you're feeling might make you feel quite ashamed, so you sort of pick and choose what to share."* (P19, 21, Female)

Participants understood that, nevertheless, effective disclosure required a certain amount of openness. Some found that the best course of action was to disclose just enough information to secure the necessary support and care but not enough to risk further stigmatization of their depression.

*"Because I think it's quite dangerous if you don't talk about it. . .And you like you need help at the end of the day, and it can spiral. If you don't talk about it and you need sort of the external sort of context, whereas if you just if you don't tell anybody about it, then you´re not sort of getting support."* (P12, 25, Female)

"*Yeah, definitely. I think it depends on the person you speak about it with, cause if someone you speak with like the by the person speaking with, they have suffered from mental illness or they understand what it's like, then it's very easy to speak to them and not feel like judged or anything like that. But I do think it's helpful to speak about it instead of bottling it up and you know.*" (P15, 21, Female)

"*Um and sharing that experience with someone. I think it's definitely very helpful, and even for a lot of people, sort of talking therapies is a good way of understanding what you're going through and help help to find ways to sort of move past that stigma.*" (P19, 21, Female)

**Subtheme: Disclosure raises questions.** Many participants recognized that disclosing mental conditions to others raised questions and required further explanations.

"*. . .because I just feel like once you start explaining that, then you have to explain multiple other things that I have no interest in talking about.*" (P22, 23, Female)

Some said they were open to discussing their depression, but they were reluctant to share more than the bare minimum (i.e., avoid sharing specific information that may lead to further stigmatizing attitudes and beliefs, such as being hospitalized or taking antidepressants) because they felt uncomfortable with the possibility of being questioned further. For example, participants expressed discomfort when asked to share the causes and reasons of their depression.

"*The problem is like, once you become more open, people start to ask questions, and I really don't wanna talk about it. . .Sharing it already is enough.*" (P22, 23, Female)

"*. . . some of the things that have caused my depression are quite private. . .I think that's the main reason for me. Um, and the reason why I keep it secret. . . would be the nature of why. Why I feel like that, and rather than this rather than the symptoms themselves, I don't mind. Like, I don't think I would mind sharing that. . . Like why I'm feeling this way and that way. But I think it was the reason, the cause of it, that would make me keep it a secret, and also because they're of, sort of. . .um how it how it may be perceived in the sense that I don't I wouldn't want to. . . Like I wouldn´t feel safe disclosing it. Um. My feelings.*" (P12, 25, Female)

## Theme four: Participants' recommendations do not align with personal preferences

This theme that emerged is directly related to our research questions in that it provides a better understanding of young people's stigmatizing subjective views. It explains the majority of participants' preferences for keeping depression a secret due to stigma's past influences, despite advising others otherwise. This is in line with our research questions in that it highlights the long-lasting impacts of stigma and one's awareness of this for their own future decision-making despite personal preferences.

The majority of the participants' recommendations did not align with their personal preferences. The vast majority of participants who were interviewed expressed a preference to keep their depression a secret for the reasons described above. Despite this, the dominant advice offered by participants was to encourage the disclosure of depression.

Incongruent advice is illustrated by the following quotes:

"...advice wise I would say definitely try speak about it. I know I don´t I said I don´t really do this, but like, I think the scariest part is like going to speak about it...because I feel like with mental health, you do feel really lonely in that sense, and you feel like you're the only one going through it...Whereas if you do speak and bring awareness to issues, then you know you'll realize, ´Ohh other people do have similar problems, and I'm not the only one.´" (P28, 25, Male)

"I've had depression since I was young. I've been hospitalized—that kind of thing. Um, I have just bottled all this up over the years. Like, they have very different kind of levels of stigma, and keeping these things to myself is just easier for me. Like I can´t get hurt if I keep it to myself. But yeah, I would definitely say, like if you have someone who can understand what you're going through, I would say like definitely talk to them instead of keeping everything to yourself..." (P03, 20, Other)

"Um...I would say for sure talk to someone. Like um, anyone... if you have someone you can trust and get out how you´re feeling, do that. I know it´s always easier said than done. Um... like I should probably take my own advice, but like I´m really bad at opening up to others about all this stuff. I would guess it helps a lot. But, yeah...then again, even if I wanted to, I guess not all of us have someone to share things like this with and be so vulnerable, if you know what I mean..." (P26, 21, Male)

## Discussion

We examined young people with depression and their views on stigma and its impacts on depression disclosure and secrecy, loneliness, and social isolation.

To address our research questions, we found that stigma led to a consequential decision to maintain secrecy or engage in disclosure, and depressed young people face potential costs when they decide to either disclose or keep their depression to themselves. While the decision to keep one's depression a secret may temporarily protect against further stigma, discrimination, and judgment, it often has negative long-term consequences, including increased loneliness, social isolation, and decreased quality relationships [31]. Despite its consequences, secrecy was also surprisingly beneficial at times. It also acted as a protective mechanism and allowed individuals to take control of their lives. On the other hand, while disclosure may expose individuals to further stigma, it may provide benefits in the long run (i.e., reduce feelings of loneliness and social isolation and increase self-esteem). However, despite its benefits, disclosure proved to also be harmful, for instance, by increasing stigma, negative judgement, and a lack of understanding.

We identified four themes that reflected patterns in participants' semi-structured interviews. Themes were used to explore research questions 1–2, which aimed to better understand the subjective views of stigma among young people with depression symptoms and the influence it has on loneliness, social isolation, secrecy, and disclosure of depression. These highlighted initial views of stigma and its varying negative and positive aspects of depression's secrecy and disclosure.

*Depression secrecy*: *positive and negative aspects* focused on the extent and reasons why stigma around depression fostered a tendency towards secrecy. Secrecy had unexpectedly positive as well as negative repercussions on young people's experiences. Specifically, our findings revealed that concealing depression can have an adaptive function, in essence, to provide a sense of safety and enable young people to take control of their lives despite their depression. Research has consistently found that many people with mental illnesses have a preference to

handle problems by themselves and an unwillingness to disclose problems [52, 53], as a result of the shame around mental illness [54]. This is consistent with other research, which indicates the possibility of both positive and negative effects of disclosure [55]. Although past studies have mostly focused on older people, research suggests that people with mental health problems feel more accepted and less stigmatized when they are surrounded by others who support and understand them [56, 57]. Research has shown how crucial mental illness disclosure can be in order to create comfort, improve quality of life [32, 56], enhance psychological growth, and aid help-seeking (i.e., contribute to the recovery process) [57].

In our sample, many participants also reported that disclosing their depression was harmful, and stigma itself fostered a desire to maintain secrecy and protect themselves from the outcomes of self-disclosure. Secrecy therefore allowed individuals to avoid rejection and stigma (additional negative judgments) by others, consequently feeling even more lonely and socially isolated. Such fears limited the desire to disclose depression and led to a vicious and recurring cycle (resulting in decreased quality of life and poorer well-being). To further address our research questions, findings have consistently shown that secrecy, despite its negative long-term effects, such as loneliness and social isolation, is a means of protecting against stigma [58]. People with depression may be reluctant to acknowledge their depression because of the guilt and stigma attached to it [59–61]. Our current findings with young adults support this existing work.

*Depression disclosure*: *positive and negative aspects*—despite existing stigmas, participants also talked about some of the reasons why they decided to disclose their depression and the benefits that came from it. In past research, those who choose to reveal may do so to feel free from the burden of keeping a critical component of their identity a secret, which is a psychological strain and emotionally stressful [30, 62]. Similarly, within our findings, participants talked about disclosure as a healing process, leading to feelings of acceptance, security, enhanced care, and understanding from others, as well as a sense of freedom from talking. Our study also highlighted the risks associated with admitting to depression. This is consistent with prior research, indicating that disclosure had a negative impact on individuals' experiences because others' nonverbal cues revealed their discomfort and judgment, which exacerbated feelings of shame and guilt [57, 58, 63]. As in Theme 1 (titled *Depression secrecy*: *positive and negative aspects*), keeping depression a secret could be both purposeful and harmful. Secrecy ultimately created a vicious, recurring cycle. In Theme 2 (titled *Depression disclosure*: *positive and negative aspects*), we addressed research questions by identifying that not everyone has the option to hide their mental condition. In particular, debilitating mental health problems inherently require disclosure in order to live a life of any quality.

*The solution is selective disclosure* was discussed by participants as a compromise between secrecy and disclosure, and this is also consistent with other studies with young people [58, 64–66]. Our participants were reassured by knowing how much information to disclose in a way that would minimize follow-up questions from others. Stigmatizing attitudes, beliefs, judgments, and self-deprecation were reduced as a result of selective disclosure. Given the importance that participants placed on both the positive and negative aspects of secrecy and disclosure, it is noteworthy that prior research has found that participants who choose to selectively disclose depression to others experience more support than discrimination [64]. Participants in our study believed that the ideal course of action was to disclose just the right amount of information to obtain the necessary care and support while avoiding exposing themselves to more stigma.

Our findings address our main research questions and are consistent with previous research showing that admitting to having a mental disorder and limiting the amount of information given to others can help reduce stereotypical attitudes and beliefs around one's diagnosed

condition [67, 68]. However, prior research has also recognized that selective disclosure can result in psychological distress and negative repercussions in relationships, as well as decreased access to suitable treatment and the ability to fulfill educational and vocational goals [69, 70]; this was not the case in our study. Furthermore, earlier studies have found full disclosure to be more closely related to experiences of support and receiving treatment [69, 70]. In the current research, we found that while disclosure is an important contributor to help-seeking, it comes with a lot of caveats, such as further stigmas.

## Limitations and future directions

The present study used individuals' conceptual model drawings to develop a semi-structured discussion focused on stigma, secrecy, and disclosure, encouraging participants to determine the direction of the interview. Yet the study's findings should be considered in light of several limitations. First, we recruited only those with high depression symptoms and may not have spoken with severely depressed young people. Furthermore, there was reasonable potential for self-selection bias; some participants might have declined to take part due to social and internalized stigma concerns. Interestingly, this study reveals that there are positive and negative aspects of both secrecy and disclosure for young people exploring how they share aspects of their depression. Selective disclosure seems key, and young people may be helped by developing a better understanding of when they prefer to share their depression and when they do not. Enabling safe sharing of young people's depression information (i.e., helping young people identify safe targets with whom they can disclose) can inform stigma-targeted treatment programs and increase public awareness regarding young people's subjective stigma views and experiences aimed at reducing the stigma around depression.

## Conclusions

In summary, this study provides new evidence of how young people with depression experience stigma and its effects on disclosure and mental health secrecy. We have revealed that young people struggle with stigma and that this can prevent them from disclosing their issues, which in turn prevents them from getting help. This work highlights the need to develop interventions to encourage young people to come forward and discuss their mental health in order to receive appropriate support and treatment. We recommend young people be signposted and have access to mental health champions or nominated teachers in their schools or universities.

## Acknowledgments

We would like to thank all participants who took part in helping us collect some of the data for this study.

## Author Contributions

**Conceptualization:** Katie Prizeman, Ciara McCabe, Netta Weinstein.

**Data curation:** Katie Prizeman.

**Formal analysis:** Katie Prizeman.

**Investigation:** Katie Prizeman.

**Methodology:** Katie Prizeman, Ciara McCabe, Netta Weinstein.

**Project administration:** Katie Prizeman, Ciara McCabe, Netta Weinstein.

**Resources:** Katie Prizeman.

**Software:** Katie Prizeman.

**Supervision:** Ciara McCabe, Netta Weinstein.

**Validation:** Katie Prizeman, Ciara McCabe, Netta Weinstein.

**Visualization:** Katie Prizeman, Ciara McCabe, Netta Weinstein.

**Writing – original draft:** Katie Prizeman.

**Writing – review & editing:** Katie Prizeman, Ciara McCabe, Netta Weinstein.

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
