## [Decision Letter · Decision Letter 0]

27 Nov 2023

PONE-D-23-29895Stigma and its Impact on Disclosure and Mental Health Secrecy in Youth with Depression Symptomology: A Qualitative AnalysisPLOS ONE

Dear Dr. Prizeman,

Thank you for submitting your manuscript to PLOS ONE. After careful consideration, we feel that it has merit but does not fully meet PLOS ONE’s publication criteria as it currently stands. Therefore, we invite you to submit a revised version of the manuscript that addresses the points raised during the review process.

We look forward to receiving your revised manuscript.

Kind regards,

Muhammad Arsyad Subu, Ph.D

Academic Editor

PLOS ONE

Journal Requirements:

“NW time on the project was funded by the European Research Council (ERC SOAR-851890). This project received no other funding.”

Please include your amended statements within your cover letter; we will change the online submission form on your behalf."

Reviewers' comments:

Reviewer's Responses to Questions

**Comments to the Author**

1. Is the manuscript technically sound, and do the data support the conclusions?

Reviewer #1: Partly

Reviewer #2: Yes

2. Has the statistical analysis been performed appropriately and rigorously? 

Reviewer #1: No

Reviewer #2: N/A

3. Have the authors made all data underlying the findings in their manuscript fully available?

Reviewer #1: Yes

Reviewer #2: Yes

4. Is the manuscript presented in an intelligible fashion and written in standard English?

Reviewer #1: Yes

Reviewer #2: Yes

5. Review Comments to the Author

Reviewer #1: 1.The term youth was used in the title, and young people were used mostly throughout the manuscript. Author should consider either one of these terms (due to different age range these terms hold)

2. In the Recruitment section, it is advised to explain how many subjects participated in the beginning, and how many were excluded due to low score MFQ.

3. Either/not participants are taking medications would also affect their communication, mood and feelings.

4. Despite of being a qualitative study, the MFQ score should be analyzed in relation to the themes (and sub-themes)

Reviewer #2: Please revise as feedback, particularly in methods, result and discussion. In methods please add the major questionnaire, in result please make it connected to the research questions and in the discussion, please check citetation style.

6. PLOS authors have the option to publish the peer review history of their article (what does this mean?). If published, this will include your full peer review and any attached files.

Reviewer #1: No

Reviewer #2: No

---

## [Author Response · Author response to Decision Letter 0]

5 Dec 2023

Reviewer's Responses to Questions:

Comments to the Author

1. Is the manuscript technically sound, and do the data support the conclusions?

Reviewer #1: Partly

Reviewer #2: Yes

2. Has the statistical analysis been performed appropriately and rigorously?

Reviewer #1: No

Reviewer #2: N/A

3. Have the authors made all data underlying the findings in their manuscript fully available?

Reviewer #1: Yes

Reviewer #2: Yes

4. Is the manuscript presented in an intelligible fashion and written in standard English?

Reviewer #1: Yes

Reviewer #2: Yes

5. Review Comments to the Author:

Reviewer #1: 

1. The term youth was used in the title, and young people were used mostly throughout the manuscript. Author should consider either one of these terms (due to different age range these terms hold)

• Response. Thank you very much for your positive feedback and thoughtful comments about our paper. This is a good point and we do agree with the reviewer and value the feedback regarding our term use. To address it we have changed the term `youth` to `young people` throughout the paper.

2. In the Recruitment section, it is advised to explain how many subjects participated in the beginning, and how many were excluded due to low score MFQ.

• Response. We have now included this information in Method section (p. 6). 

3. Either/not participants are taking medications would also affect their communication, mood and feelings.

• Response. We take the reviewer’s point regarding medication taking of our participants. That said, we did not collect data on this information. To address this, we have included a sentence in the Participants and Recruitment section: “Nor did we collect data on participant psychiatric medication.” (p. 8)

4. Despite of being a qualitative study, the MFQ score should be analyzed in relation to the themes (and sub-themes)

• Response. We recruited only MFQ scores over 27, which is considered the clinical cut-off for high depression symptoms. Therefore, it is not clear to us how we would further delineate between depression symptoms in this sample. Further, as we only have 28 participants (due to qualitative data saturation), we have no statistical power to do a quantitative analysis between the MFQ data and the sub-themes. 

This being said, to address the reviewer's point, we have included the following information in the results section: "Participants' demographics and clinical characteristics are presented in Table 2. All participants had an MFQ of <27. A cut-off score of 27 and above has been identified as the difference between clinical and non-clinical levels of depressive symptoms (44)." (p. 12) 

We have also moved Table 2: Participants' demographics and clinical characteristics to the results section (p. 13).

Reviewer #2: 

1. Please revise as feedback, particularly in methods, result and discussion. In methods please add the major questionnaire, in result please make it connected to the research questions and in the discussion, please check citetation style.

• Response. Thank you very much for your constructive feedback about our paper. We have included and referred to our major questionnaire in the methods section (Table 1) (p. 6-7). 

• Response. We have included information in the results section to connect it to research questions. “Four interrelated broad themes emerged from the data. Themes were used to explore research questions 1–2, which aimed to better understand the subjective views of stigma among young people with depression symptoms and the influence it has on loneliness, social isolation, secrecy, and disclosure of depression.” (p. 12). 

We have also included information throughout the result (p. 12-29) and discussion (p. 29-33) sections to better connect each theme back to our research questions. 

• Response. We have amended any citation style mistakes throughout the paper. 

6. PLOS authors have the option to publish the peer review history of their article (what does this mean?). If published, this will include your full peer review and any attached files.

Do you want your identity to be public for this peer review? For information about this choice, including consent withdrawal, please see our Privacy Policy.

Reviewer #1: No

Reviewer #2: No

---

## [Editor Report · Decision Letter 1]

8 Dec 2023

Stigma and its impact on disclosure and mental health secrecy in young people with depression symptomology: a qualitative analysis

PONE-D-23-29895R1

Dear Dr. Prizeman,

We’re pleased to inform you that your manuscript has been judged scientifically suitable for publication and will be formally accepted for publication once it meets all outstanding technical requirements.

Kind regards,

Muhammad Arsyad Subu, Ph.D

Academic Editor

PLOS ONE
---

## [Editor Report · Acceptance letter]

21 Dec 2023

PONE-D-23-29895R1 

PLOS ONE

Dear Dr. Prizeman, 

I'm pleased to inform you that your manuscript has been deemed suitable for publication in PLOS ONE. Congratulations! Your manuscript is now being handed over to our production team.

Kind regards, 

on behalf of

Dr. Muhammad Arsyad Subu 

Academic Editor

PLOS ONE